# Prevalence and associated factors of delayed first antenatal care booking among reproductive age women in Ethiopia; a multilevel analysis of EDHS 2016 data

Achamyeleh Birhanu Teshale * , Getayeneh Antehunegn Tesema

Department of Epidemiology and Biostatistics, Institute of Public Health, College of Medicine and Health Sciences, University of Gondar, Gondar, Ethiopia

* achambir08@gmail.com

## Abstract

### Background

Early or timely initiation of antenatal care and regular visits based on the schedule have a tremendous effect on both maternal and fetal health. Despite the paramount benefits of early initiation of ANC within the first 12 weeks of pregnancy, women still do not have adequate and equal access to high-quality early antenatal care.

### Objective

To determine the prevalence and factors associated with delayed first ANC booking in Ethiopia.

### Method

A secondary data analysis was conducted using the 2016 Ethiopian demographic and health survey data. All reproductive age women who gave birth in the five years preceding the survey and who had ANC visit for their last child were included in this study. The total weighted sample size analyzed was 4,741. Due to the hierarchical nature of the EDHS data, Multi-level logistic regression model was used to identify the individual and community level factors associated with delayed first ANC booking.

### Result

In this study, the prevalence of delayed first ANC booking was 67.31% [95% CI: 65.96% to 68.63%]. Women with secondary and higher education [Adjusted Odd Ratio (AOR) = 0.78; 95%CI: 0.61, 0.99] and [AOR = 0.61; 95%CI: 0.44, 0.83] respectively had lower odds of delayed first ANC booking. But woman who were multiparous and grand multiparous [AOR = 1.21; 95%CI: 1.01, 1.45] and [AOR = 1.50; 95%CI: 1.16, 1.93] respectively, women with the last pregnancy wanted no more [AOR = 1.52; 95%CI: 1.10, 2.09], a woman who was living in the rural area [AOR = 1.66; 95%CI: 1.25, 2.21], and a woman who was living in large

**Data Availability Statement:** All relevant result-based data are available within the manuscript and anyone can access the data set online from www.measuredhs.com.

**Funding:** The author(s) received no specific funding for this work.

**Competing interests:** The authors have declared that no competing interests exist.

**Abbreviations:** ANC, Antenatal Care; DHS, Demographic and Health Survey; EDHS, Ethiopian Demographic and Health Survey; ICC, Intra-class Correlation Coefficient; MOR, Median Odds Ratio; PCV, Proportional Change in Variance; PHS, Population and Housing Census; WHO, World Health Organization.

central regions and small peripheral regions [AOR = 2.76; 95%CI: 2.20, 3.47] and [AOR = 2.70; 95%CI: 2.12, 3.45] respectively had higher odds of delayed first ANC booking.

## Conclusion

Despite the documented benefits of early antenatal care initiation, late ANC booking is still predominant in Ethiopia as highlighted by this study. Maternal education, parity, wanted the last child, residence and region were significantly associated with delayed first ANC booking. Therefore, taking special attention for these high-risk groups could decrease delayed first ANC booking and this intern decreases maternal and fetal health problems by identifying and intervene early.

## Background

Maternal health is defined as the health of the women before pregnancy, during pregnancy, during childbirth and the postpartum period [1]. Worldwide, maternal mortality was dropped by 44% between the years 1990 and 2015, in which 808 women (99% were from developing countries especially in Sub-Saharan Africa) die every day from preventable causes related to pregnancy and childbirth [2, 3]. Based on the World Health Organization (WHO) report, the maternal mortality ratio in Ethiopia was decreased from 1,250 in 1990 to 353 in 2015, which is sluggish progress [3]. Even though the Sustainable Development Goal targeted to reduce the global maternal mortality ratio to less than 70 per 100 000 live births between 2016 and 2030, still it is difficult to achieve this agenda with the current progress [1]. So, appropriate and timely Utilization of maternal health services will improve pregnancy outcomes and reduce the death of the mother and the neonate [4–8].

Improving maternal health is one of the WHO key priorities and antenatal care (ANC) is one of the four pillars of the initiative for safe motherhood [9–11]. Antenatal care during pregnancy is an entry point for pregnant women into the health care system and offers an opportunity to organize the necessary services for ensuring a healthy pregnancy, safe delivery and a healthy mother-baby pair [12, 13].

The WHO recommends a minimum of four ANC contacts to reduce maternal and perinatal mortality with the first visit at a gestational age within 12 weeks [14, 15]. Early or timely initiation of ANC and regular visits based on the schedule have a tremendous effect on both maternal and fetal health [14, 16]. Besides, early ANC visit allows health care providers to screen and treat different maternal and fetal health problems such as malnutrition, sexually transmitted diseases, congenital anomalies, and other pregnancy-related complications as early as possible [9, 17–19]

In spite of all these aforementioned benefits of early initiation of ANC visit and the WHO recommendation of the first ANC visit be within the first 12 weeks of pregnancy, women still do not have adequate and equal access to the high-quality early antenatal care [20–22]. Even though the coverage of early antenatal care visit in the world was increased by 43% between 1990 and 2013, only less than half of all women in less developed countries had early antenatal care visit by 2013 [22]. Specifically, in Africa, late ANC booking is still a devastating issue which ranges from 44% in Cameroon and 86.6% in Zambia [10, 23–28]. Ethiopia is also one of the developing African countries with a high prevalence of late ANC initiation which ranges from 42% in Addis Ababa to 81.8% in western Wolega [29–39].

Different studies across the world revealed factors like maternal education [23, 27, 40, 41], maternal age [23, 27, 32, 34, 37], pregnancy intention [23, 29, 36, 40, 42], wealth status/income [28, 34], parity [10, 25, 29, 40, 42], having history of abortion or still birth [10], marital status [24, 25, 42], exposure to mass media [41, 43], distance from the health facility [42, 43], residence [40, 41], health insurance [42, 43] and region [41, 44] are associated with delayed first ANC booking.

Most of the previous studies in Ethiopia, on the timely initiation of ANC visits, are institutionally-based and restricted to specific regions or zones and with small sample size. But this study seeks to use a nationally representative data to determine the prevalence and factors associated with delayed first ANC booking and the findings of this study will help policy-makers in the implementation of interventions that will increase the timely initiation of ANC visit and contribute to the promotion of maternal and fetal/neonatal health in Ethiopia.

## Method

### Study setting

The study was conducted in Ethiopia which is ethno-linguistically classified into nine regional states, two chartered cities, 611 weredas, and 15,000 Kebeles. The regions are administratively divided into zones and zones into woredas (which is the third administrative divisions of Ethiopia). Finally, at level four, woredas are further subdivided into the lowest administrative unit called kebele. Next to Nigeria (with over 180 million people), Ethiopia is the most populous country in Africa with an estimated 100 million people. Almost half of the population of Ethiopia are females with 21% in their reproductive ages [45]. Regarding the health care system in Ethiopia, the fourth health sector development plan introduced a three-tier health-delivery service system. The primary health care unities (health posts and health centers) and primary hospitals are at primary level, secondary level services are given by general hospitals, and tertiary services are delivered by specialized hospitals [46].

### Data source, study population and sampling technique

This study was based on the Ethiopian demographic and health survey (EDHS) 2016 data which was a nationally representative sample conducted from January 18, 2016, to June 27, 2016. To select enumeration areas for EDHS 2016, a total of 84,915 Enumeration areas (EAs) from an Ethiopian Population and Housing Census (PHC) conducted in 2007 were used as a sampling frame. Regarding the sampling technique, the survey used a two-stage stratified cluster sampling technique selected in two stages. In the first stage a total of 645 EAs (443 in rural areas) and in the second stage an average of 28 households per each cluster were selected. Any further information about the data/survey exists in the 2016 EDHS report [47].

For this study, we have used the kids' data set and the study population was women (aged 15 to 49 years) who gave birth five years preceding the survey and attended antenatal care during their last pregnancy. Among women with two and above live births in the previous five years, the information taken correspond to the latest birth.

### Study variables

The outcome variable for this study was delayed first ANC booking which is defined as booking first ANC after 12 weeks of gestation [15]. In this study, both individual and community-level independent variables were considered. The individual-level factors included were; maternal age, marital status, maternal educational level, household wealth, media exposure, insurance coverage, parity, ever had of a terminated pregnancy, and pregnancy intention. The

four variables, place of residence, region, perception of distance from the health facility and community level media exposure were considered as community-level factors. The community-level variable, community-level media exposure, was obtained by aggregating the individual level media exposure into clusters by using the proportion of those who had media exposure and this community-level media exposure shows the overall media exposure in the community. Median values were used to categorize as high and low because the aggregated variable had skewed distribution. In this study region was recategorized into three categories; larger central [Tigray, Amhara, Oromia, and Sothern Nations Nationalities and Peoples Region], small peripherals [Afar, Somali, Benishangul, and Gambela], and metropolis[Harari, Dire Dawa, and Addis Ababa] based on their geopolitical features, consistent with a previous study from Ethiopia [48, 49] (Fig 1).

## Data management and statistical analysis

Data were extracted from EDHS 2016 and further coding and analysis were done using Stata version 14. Throughout analysis sample weights were done to adjust for non-proportional allocation of the sample to strata and regions during the survey process and to restore the representativeness. A multi-level logistic regression analysis was used to account for the hierarchal nature of the DHS data. First bivariable multilevel logistic regression analysis was performed and those variables with p-value <0.20 were considered for multivariable analysis.

After selecting variables for multivariable analysis, four models; the null model (without explanatory variables), model 1 (containing only individual-level factors), model 2 (examined the effect of community-level factors) and model 3 (which incorporate both individual and community level factors) were fitted. Since these models were nested, deviance was used to assess the model fitness and the model with lower deviance (model 3) was the best-fitted model. In addition, multicollinearity was tested using the variance inflation factor (VIF) and we have got a VIF of less than five for each independent variable with a mean VIF of 1.89, indicating there was no significant multicollinearity between independent variables.

The random effects (the amount of community variation), which are measures of variation of delayed first ANC visit across communities or clusters, were expressed in terms of the Intra-Class Correlation (ICC), median odds ratio(MOR) and proportional change in variance (PCV) [50–53]. These ICC, MOR, and PCV were calculated to quantify; the degree of homogeneity of delayed first ANC booking within clusters, the degree of variation of delayed first ANC booking across clusters in terms of odds ratio scale, and the proportion of variance explained by consecutive models respectively.

## Ethical consideration

This study is a secondary analysis of the 2016 EDHS data, so it does not require ethical approval. For conducting our study, we registered and requested the dataset from DHS on-line archive and received approval to access and download the data files. According to the EDHS 2016 report, all participant data were anonymized during the collection of the survey data [47].

## Result

### Background characteristics of study participants

Data from a weighted sample of 4,741 women aged 15–49 years who gave birth in the five years preceding the survey and who attend ANC visit for their last pregnancy were included in this analysis. The median age of the participants was 28(±9) years. In general, the majority

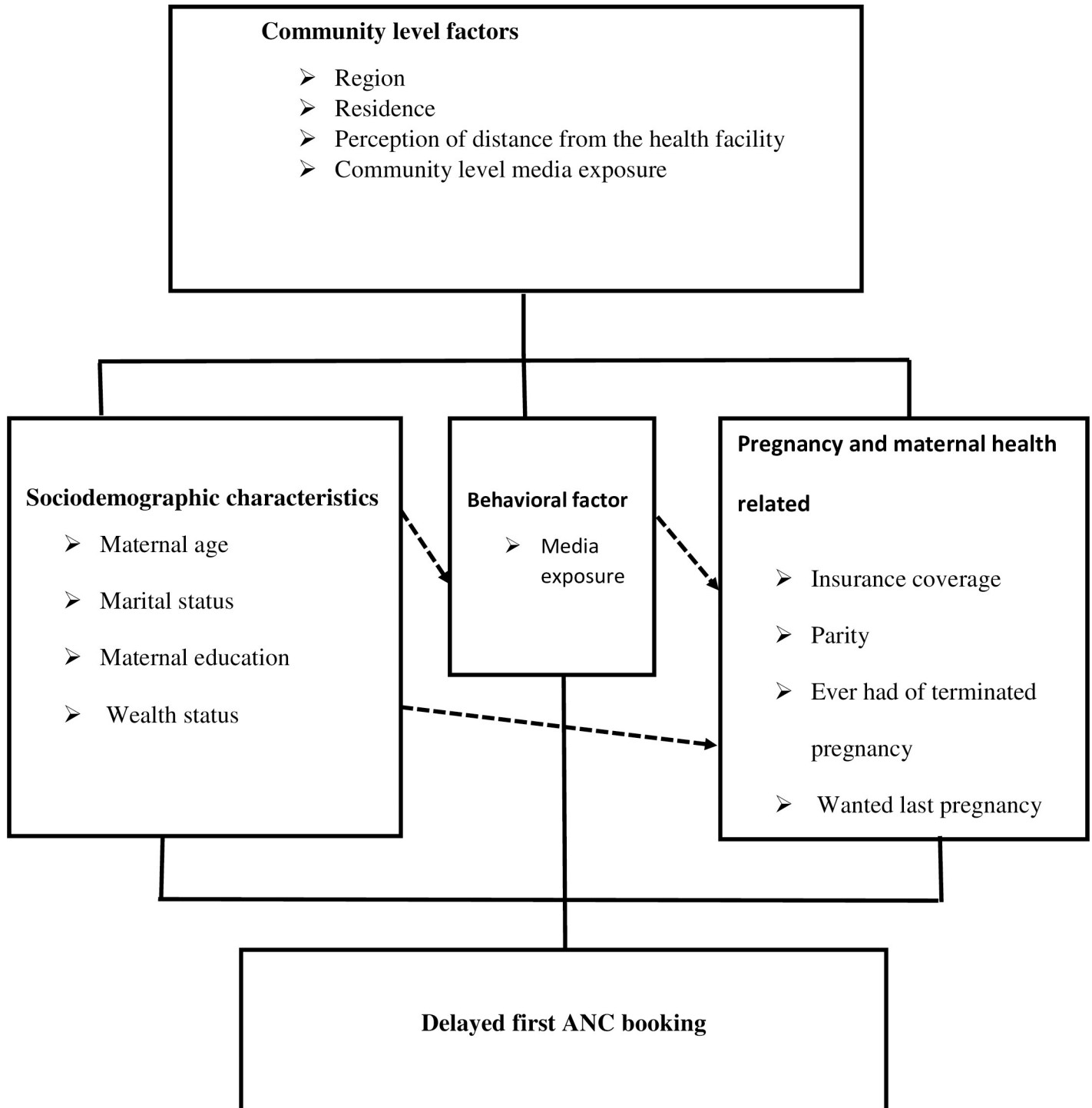

**Fig 1. Conceptual framework of factors associated with delayed first ANC visit developed from searching of literature.**

(52.49%) of women were aged between 25 and 34 years, 53.49% of them had no formal education and 93.87% of participants were married. Moreover, 81.56% of women were rural dwellers and 90.34% were from large central regions. Around half (50.30%) of the women were living

**Table 1. Background characteristics of study participants.**

| Respondent characteristics | Frequency | Percentage |
|---|---|---|
| Maternal age(years) | | |
| 15–24 | 1,225 | 25.84 |
| 25–34 | 2,489 | 52.49 |
| 35–49 | 1,027 | 21.66 |
| Maternal education | | |
| No formal education | 2,556 | 53.90 |
| Primary | 1,572 | 33.15 |
| Secondary | 387 | 8.16 |
| Higher education | 226 | 4.79 |
| Marital status | | |
| Married | 4,450 | 93.87 |
| Not married | 291 | 6.13 |
| Household wealth | | |
| Poorest | 787 | 16.60 |
| Poorer | 932 | 19.65 |
| Middle | 984 | 20.75 |
| Richer | 963 | 20.32 |
| Richest | 1,075 | 22.68 |
| Media exposure | | |
| Yes | 2,722 | 57.41 |
| No | 2,019 | 42.59 |
| Insurance coverage | | |
| Yes | 238 | 5.02 |
| No | 4,503 | 94.98 |
| Parity | | |
| Primiparous | 1,112 | 23.45 |
| Multiparous | 2,084 | 43.95 |
| Grand multiparous | 1,545 | 32.60 |
| Ever had of a terminated pregnancy | | |
| Yes | 436 | 90.82 |
| No | 4,305 | 9.18 |
| Wanted last pregnancy | | |
| Wanted then | 3,605 | 76.04 |
| Wanted later | 820 | 17.29 |
| Wanted no more | 316 | 6.67 |
| Region | | |
| Metropolitan | 233 | 4.91 |
| Large central | 4,283 | 90.34 |
| Small peripheral | 225 | 4.75 |
| Residence | | |
| Urban | 874 | 18.44 |
| Rural | 3,867 | 81.56 |
| Community-level Media exposure | | |
| Lower | 2,385 | 50.30 |
| Higher | 2,356 | 49.70 |
| Distance from the health facility | | |
| Big problem | 2,383 | 50.26 |
| Not a big problem | 2,358 | 49.74 |

in communities with a higher proportion of media exposure and 50.26% of women perceive distance from the health facility as a big problem (Table 1).

## Prevalence of delayed first ANC booking in Ethiopia

In this study the prevalence of early and delayed ANC attendance was 32.69% (31.37–34.04) and 67.31% (65.96–68.63) respectively. Of those who had delayed ANC booking, the majority (41.60%) of study participants had their first antenatal care visit at 4 to 5 months of gestation (Fig 2).

## Random effects and model fitness

Table 2 revealed that in the null model, about 18.5% of the total variation on delayed first ANC booking was occurred at the cluster level and is attributable to the community-level factors. In addition, the null model also had the highest MOR value (2.27) indicating when randomly select an individual from one cluster with a higher risk of delayed first ANC booking and the other cluster at lower risk, individuals at the cluster with a higher risk of delayed first ANC booking had 2.27 times higher odds of having a delayed first ANC booking as compared with their counterparts. Furthermore, the highest (64.1%) PCV in the full model (model 3), indicates that 64.1% of the community-level variation on delayed first ANC booking was explained by the combined factors at both the individual and community levels. The model fitness was done using deviance in which the final model (model 3) was the best-fitted model since it had the lowest deviance (5,746.14).

## Individual and community-level factors associated with delayed first ANC booking

In the bivariable multilevel modeling, all of the explanatory variables (both individual level and community level variables) except ever had of a terminated pregnancy had shown statistically significant association at a p-value of <0.20.

In multivariable multilevel logistic regression analysis, where both the individual and community level factors were fitted simultaneously; maternal education, parity, wanted last pregnancy, residence and region were significantly associated with delayed first ANC booking.

The odd of delayed first ANC booking was 22% [adjusted odds ratio (AOR) = 0.78; 95% CI: 0.61, 0.99] and 39% [AOR = 0.61; 95%CI: 0.44, 0.83] lower in mothers who had secondary education and higher education respectively as compared to those mothers who had no formal education. The woman who were multiparous and grand multiparous had 1.21 times [AOR = 1.21; 95%CI: 1.01, 1.45] and 1.50 times [AOR = 1.50; 95%CI: 1.16, 1.93] higher odds of delayed ANC booking as compared to primiparous women. Regarding pregnancy intention/ wanted last pregnancy, women with the last pregnancy wanted no more had 1.52 times [AOR = 1.52; 95%CI: 1.10, 2.09] higher odds of delayed first ANC visit as compared to mothers with wanted last pregnancy then.

A woman who was living in the rural area had 1.66 [AOR = 1.66; 95%CI: 1.25, 2.21] times higher odds of delayed first ANC booking as compared with a woman who was living in urban areas. Regarding region, a woman who was living in the large central and small peripheral regions had 2.76 times [AOR = 2.76; 95%CI: 2.20, 3.47] and 2.70 times [AOR = 2.70; 95%CI: 2.12, 3.45] higher odds of delayed first ANC booking respectively, as compared to a woman from the metropolitan regions (Table 3).

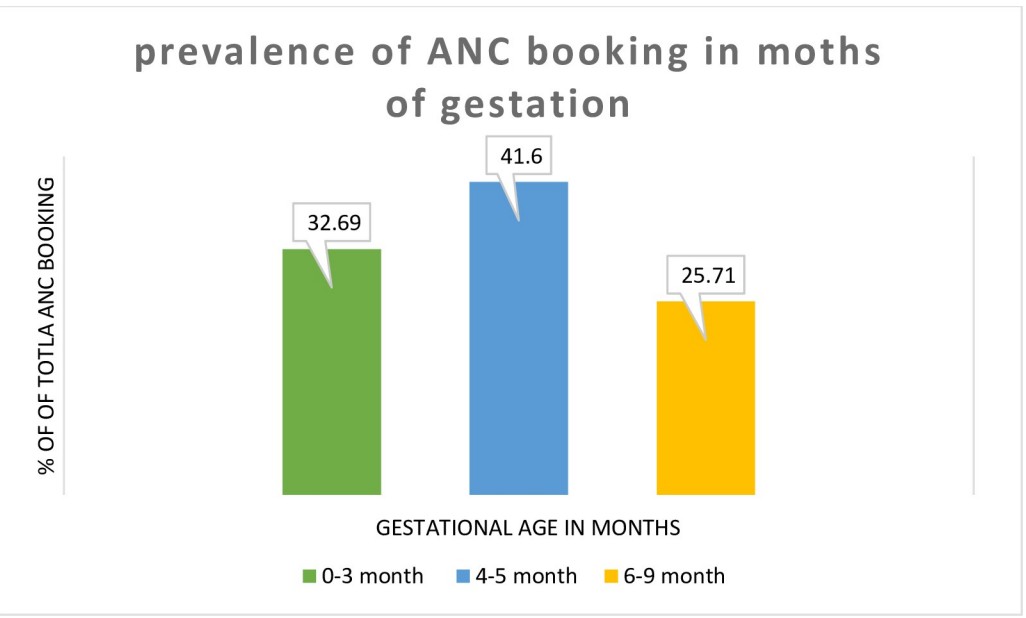

**Fig 2. Prevalence of delayed first ANC booking in Ethiopia, 2016.**

## Discussion

The study attempted to assess the prevalence and associated factors of delayed first ANC booking in Ethiopia. This study reported 67.31% of pregnant women had booked their first ANC late. This finding is consistent with studies done in the Kembata Tembaro zone, Hadiya zone and a study done using EDHS 2011 [31, 38, 39]. This proportion of pregnant women who booked late was found lower compared to different studies conducted in Africa and Ethiopia [10, 23–27, 29, 34, 35] and higher than different studies conducted in different countries [28, 30, 32, 33, 36, 37]. The discrepancy might be due to most of the indicated studies are institution-based with small sample size. The way they operationalizing the outcome variable (delayed first ANC booking) might also be the reason for the discrepancy because most of studies having a lower proportion of delayed first ANC booking classified mothers as delayed for booking first ANC if they come after 16 weeks of gestation, while our study classified a mother as being late if she came after 12 weeks. Besides, the discrepancy of this finding with that of the findings of studies conducted out of Ethiopia might be due to socio-demographic and cultural differences.

In this study, we observed that women with secondary and higher education are less likely to have delayed first ANC booking than women with no formal education. This is supported by studies done in Nigeria [27, 41], Myanmar [40] and Zambia [23] which similarly showed

**Table 2. Multilevel parameters showing random effects on delayed first ANC booking and model fitness.**

| Parameter | Null model | Model 1 | Model 2 | Model 3 |
|---|---|---|---|---|
| Community level variance (SE) | 0.746(0.096) | 0.424(0.070) | 0.285(0.058) | 0.268(0.058) |
| MOR | 2.273 | 1.857 | 1.660 | 1.636 |
| PCV | Ref | 0.433 | 0.618 | 0.641 |
| ICC | 0.185 | 0.114 | 0.080 | 0.075 |
| Deviance | 6,044.76 | 5,858.82 | 5,808.46 | 5,746.14 |

**Table 3. Multilevel multivariable analysis of factors associated with delayed first ANC booking in Ethiopia, EDHS 2016.**

| Respondent characteristics | Null model | Model 1 (AOR 95%CI) | Model 2 (AOR 95%CI) | Model 3 (AOR 95%CI) |
|---|---|---|---|---|
| *Individual-level and household level factors* | | | | |
| Maternal age(years) | | | | |
| 15–24 | | 1.00 | | 1.00 |
| 25–34 | | 0.91(0.76–1.10) | | 0.98(0.82–1.18) |
| 35–49 | | 0.86(0.67–1.12) | | 0.96(0.75–1.24) |
| Maternal education | | | | |
| No formal education | | 1.00 | | 1.00 |
| Primary | | 0.86(0.76–1.02) | | 0.90(0.76–1.06) |
| Secondary | | 0.71(0.56–0.91) | | 0.78(0.61–0.99) |
| Higher education | | 0.53(0.39–0.73) | | 0.61(0.44–0.83) |
| Marital status | | | | |
| Not married | | 1.00 | | 1.00 |
| Married | | 1.19(0.93–1.52) | | 1.13(0.88–1.44) |
| Household wealth | | | | |
| Poorest | | 1.00 | | 1.00 |
| Poorer | | 1.04(0.83–1.31) | | 1.05(0.83–1.31) |
| Middle | | 0.85(0.67–1.08) | | 0.85(0.67–1.07) |
| Richer | | 0.93(0.72–1.19) | | 0.92(0.72–1.18) |
| Richest | | 0.52(0.41–0.66) | | 0.95(0.70–1.29) |
| Media exposure | | | | |
| No | | 1.00 | | 1.00 |
| Yes | | 0.90(0.76–1.06) | | 0.93(0.78–1.11) |
| Insurance cover | | | | |
| Yes | | 0.80(0.58–1.12) | | 0.75(0.54–1.04) |
| No | | 1.00 | | 1.00 |
| Parity | | | | |
| Primiparous | | 1.00 | | 1.00 |
| Multiparous | | 1.26(0.92–1.35) | | 1.21(1.01–1.45) |
| Grand multiparous | | 1.63(1.26–2.10) | | *1.50(1.16–1.93)* |
| Wanted last pregnancy | | | | |
| Wanted then | | 1.00 | | 1.00 |
| Wanted later | | 1.12(0.92–1.35) | | 1.13(0.94–1.37) |
| Wanted no more | | 1.49(1.08–2.05) | | 1.52(1.10–2.09) |
| *Community-level factors* | | | | |
| Residence | | | | |
| Urban | | | 1.00 | 1.00 |
| Rural | | | 2.01(1.61–2.51) | *1.66(1.25–2.21)* |
| Region | | | | |
| Metropolitan | | | 1.00 | 1.00 |
| Large central | | | 2.80(2.23–351) | 2.76(2.20–3.47) |
| Small peripheral | | | 2.85(2.25–3.62) | 2.70(2.12–3.45) |
| Community-level Media exposure | | | | |
| Lower | | | 1.00 | 1.00 |
| Higher | | | 1.03(0.85–1.24) | 1.11(0.91–1.37) |
| Distance from the health facility | | | | |
| Big problem | | | 1.00 | 1.00 |
| Not a big problem | | | 0.97(0.84–1.13) | 1.02(0.88–1.18) |

that women with higher education were less likely for delayed first ANC booking. This might be due to the levels and ways of understanding regarding the negative effect of delayed ANC booking is different among women with different levels of education. That is educated women would likely appreciate or know the problems related to delayed first ANC booking more than those who had no formal education.

Parity was a factor for late booking for ANC in which multiparous and grand multiparous mothers were more likely to book late for the first ANC compared to those with primiparous. This finding was similar to those of studies done in Debrebirhan-Ethiopia, Zambia, Myanmar, Rwanda and Tanzania [10, 25, 29, 40, 42]. This might be due to women with more pregnancies previously do not want to start ANC early because they already know what things will happen during pregnancy and childbirth. In addition, they may also find it harder to attend ANC early because of the burden of childcare or they are too busy in caring a larger family to book early. The other possible explanation might be health education received in their previous pregnancies might be ineffective in changing or modifying their behaviors.

In this study, Pregnant mothers with unplanned pregnancy (pregnancy wanted no more) were more likely to book late for ANC than those in which the pregnancy was planned. This finding was agreed with studies done in Addis Zemen-Ethiopia, Debre-Birhan Ethiopia, Myanmar, Zambia, and Rwanda [23, 29, 36, 40, 42]. This might be due to a woman with unplanned pregnancy might have a chance of detecting the pregnancy later or the mother may give less attention and love to this pregnancy. Furthermore, a woman with unwanted pregnancy might not seek appropriate care for their pregnancy and might not be willing to get any information related to ANC from health care professionals and their peers or friends.

The fourth important finding in this study is the role of place of residence in which mothers from the rural residence were more likely to book late for the first ANC compared to their counterparts. This finding is supported by different studies in Myanmar and Nigeria [40, 41]. This is due to mothers who live in rural areas are less likely to utilize maternal health services like timely initiation of ANC because of its inadequate availability and accessibility and due to unequal distribution of health facilities as well as health personnel between the urban and rural areas.

Furthermore, in this study region is also associated with ANC booking. Mothers from large central regions and small peripheral regions were more likely to have late ANC booking as compared to metropolitans. This is congruent with a study done in Nigeria and Wales [41, 44] which showed that region is the uniform and consistent predictors of delay in ANC initiation. This might be due to inadequate and improper or unequal distribution of maternal health services, due to scarcity of resources in poor clinical settings like Ethiopia, in which most of the services are concentrated in near urban areas such as metropolitans or city administrations of Ethiopia.

The main strength of this study was that it used a nationally representative data with large sample size. The other strength was that we employed an advanced and appropriate statistical approach (multilevel analysis) to accommodate the hierarchical nature of the data. However, this study had limitations in that the EDHS survey is relied on respondents' self-report and might have the possibility of recall bias because respondents/mothers were asked to remember things happened in the past. Again, this study only generates associations between delayed first ANC booking and some important individual-level and community-level factors that is limited in its design to establish causality between the outcome of interest and these important independent variables.

## Conclusion

Despite the documented benefits of early antenatal care initiation, late ANC booking is still predominant in our country as highlighted by this study. Maternal education, parity, wanted

last pregnancy, residence and region were significantly associated with delayed first ANC booking. Therefore, intervention efforts to improve early first ANC booking in Ethiopia requires targeting of these hindering factors by taking special attention to mothers who had no formal education, multiparous and grand multiparous mothers, and mothers with an unwanted pregnancy. Moreover, it is also better to consider mothers from rural areas and mothers from regions other than metropolitans since these groups of mothers might not have access to maternal health services timely.

## Acknowledgments

We are grateful to thank the MEASURE DHS program for permitting us to obtain and use the 2016 EDHS data set.

## Author Contributions

**Conceptualization:** Achamyeleh Birhanu Teshale, Getayeneh Antehunegn Tesema.

**Data curation:** Achamyeleh Birhanu Teshale, Getayeneh Antehunegn Tesema.

**Formal analysis:** Achamyeleh Birhanu Teshale, Getayeneh Antehunegn Tesema.

**Investigation:** Achamyeleh Birhanu Teshale, Getayeneh Antehunegn Tesema.

**Methodology:** Achamyeleh Birhanu Teshale, Getayeneh Antehunegn Tesema.

**Resources:** Achamyeleh Birhanu Teshale, Getayeneh Antehunegn Tesema.

**Software:** Achamyeleh Birhanu Teshale, Getayeneh Antehunegn Tesema.

**Validation:** Achamyeleh Birhanu Teshale, Getayeneh Antehunegn Tesema.

**Visualization:** Achamyeleh Birhanu Teshale, Getayeneh Antehunegn Tesema.

**Writing – original draft:** Achamyeleh Birhanu Teshale, Getayeneh Antehunegn Tesema.

**Writing – review & editing:** Achamyeleh Birhanu Teshale, Getayeneh Antehunegn Tesema.

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
