## [Decision Letter · Decision Letter 0]

4 Mar 2020

PONE-D-19-33776

Prevalence and associated factors of delayed first Antenatal care booking among reproductive age women in Ethiopia; a multilevel analysis of EDHS 2016 data.

PLOS ONE

Dear Mr. Teshale,

Thank you for submitting your manuscript to PLOS ONE. After careful consideration, we feel that it has merit but does not fully meet PLOS ONE’s publication criteria as it currently stands. Therefore, we invite you to submit a revised version of the manuscript that addresses the points raised during the review process.

We would appreciate receiving your revised manuscript by Apr 18 2020 11:59PM. To enhance the reproducibility of your results, we recommend that if applicable you deposit your laboratory protocols in protocols.io, where a protocol can be assigned its own identifier (DOI) such that it can be cited independently in the future. For instructions see: http://journals.plos.org/plosone/s/submission-guidelines#loc-laboratory-protocols

We look forward to receiving your revised manuscript.

Kind regards,

Joshua Amo-Adjei, Ph.D

Academic Editor

PLOS ONE

Additional Editor Comments (if provided):

As you (authors) would notice, the second reviewer makes very important technical comments that should be properly addressed before the manuscript can be considered publishable.

Journal Requirements:

2. In ethics statement in the manuscript and in the online submission form, please confirm whether all participant data were fully anonymized before you accessed them.

3. We noticed you have some minor occurrence(s) of overlapping text with the following previous publication(s), which needs to be addressed:

https://doi.org/10.1186/s12905-018-0690-1

https://doi.org/10.3390/ijerph16050748

https://doi.org/10.1186/s12889-019-6845-7

https://www.who.int/reproductivehealth/early-anc-worldwide/en/

In your revision ensure you cite all your sources (including your own works), and quote or rephrase any duplicated text outside the Methods section. Further consideration is dependent on these concerns being addressed.

4. Your ethics statement must appear in the Methods section of your manuscript. If your ethics statement is written in any section besides the Methods, please move it to the Methods section and delete it from any other section. Please also ensure that your ethics statement is included in your manuscript, as the ethics section of your online submission will not be published alongside your manuscript.

Reviewers' comments:

Reviewer's Responses to Questions

**Comments to the Author**

1. Is the manuscript technically sound, and do the data support the conclusions?

Reviewer #1: Yes

Reviewer #2: Partly

2. Has the statistical analysis been performed appropriately and rigorously? 

Reviewer #1: Yes

Reviewer #2: No

3. Have the authors made all data underlying the findings in their manuscript fully available?

Reviewer #1: Yes

Reviewer #2: Yes

4. Is the manuscript presented in an intelligible fashion and written in standard English?

Reviewer #1: Yes

Reviewer #2: Yes

5. Review Comments to the Author

Reviewer #1: The manuscript is well written. The analysis is well done. There are some few observations.

Background

Line 74 to 76 needs a reference.

The sentence starting from line 76 to 79 is not clear and need to be revised

Methods

Study setting

A brief discussion about the health care system in Ethiopia who give the reader insight.

Data source and study population

Line 117, the authors said the used the kid’s data set of EDHS 2016… my question why did you not use the women’s dataset and chose the kid’s dataset?

Results

Line 174 to 176 you must be consistent with decimal places

Table 1 there were no values for parity

Line 177 to 180 needs to be revised. It is not clear

Discussion

This section was well written and conclusions were drawn from the results

The manuscript should be proof read

Reviewer #2: Lines 107-109. The author refers to the Ethiopian Census 2007 as the sampling frame of the EDHS 2016. This may not be very correct as DHS surveys only use the Census data to select Enumeration Areas. Then, a listing exercise is conducted in each selected EA to update the list of households. The updated list of hhds is used to select households to interview. The authors could revise the statement according to this methodology.

A conceptual framework to help readers understand why these variables were included in the study should have supported the description of the study variables

Lines 145-147. Sample weights should be used throughout the analyses, not only for frequencies and for proportions. Could the author clarify whether the multilevel analysis was weighted or not.

Were collinearity checks performed on the independent variables before ruining MV analyses?

The paragraph on the random effects does not add value to the paper; it rather brings statistical information that the author seems struggling to explain properly. The author uses values (va, π) that are not documented in the paper. Furthermore, I would expect the author to first clarify why a random effect model was needed on these data and then explain what those effects (if any) could be referring to in practice.

Presenting the results section by random vs. fixed effects seems to turn the paper to a methodological paper, comparing methods, rather than keeping the focus on describing the factors associated with delayed ANC. In addition, a number of statements in this section are subjective interpretations of statistical models which are not the focus of the paper and not expected in a result section. This section should be entirely rewritten to clearly highly the key results of the analysis that would be discussed later.

6. PLOS authors have the option to publish the peer review history of their article (what does this mean?). If published, this will include your full peer review and any attached files.

Reviewer #2: No

---

## [Author Response · Author response to Decision Letter 0]

13 Mar 2020

Dear Editor thank you for your important comments, here below are the authors responses 

Authors response; we followed the PLOS ONE's style while writing our revised manuscript 

2. In ethics statement in the manuscript and in the online submission form, please confirm whether all participant data were fully anonymized before you accessed them.

Authors response; the author amended the ethical statement accordingly in the revised manuscript

3. We noticed you have some minor occurrence(s) of overlapping text with the following previous publication(s), which needs to be addressed:

https://doi.org/10.1186/s12905-018-0690-1

https://doi.org/10.3390/ijerph16050748

https://doi.org/10.1186/s12889-019-6845-7

https://www.who.int/reproductivehealth/early-anc-worldwide/en/

In your revision ensure you cite all your sources (including your own works), and quote or rephrase any duplicated text outside the Methods section. Further consideration is dependent on these concerns being addressed.

Authors response; after checking the overlapping texts with given papers, the overlapped texts are re written up or rephrased accordingly in the revised manuscript.

4. Your ethics statement must appear in the Methods section of your manuscript. If your ethics statement is written in any section besides the Methods, please move it to the Methods section and delete it from any other section. Please also ensure that your ethics statement is included in your manuscript, as the ethics section of your online submission will not be published alongside your manuscript.

Authors response; the ethical statement was in the manuscript, method part, and in the revised manuscript the ethical statement also exists in the methods section of the manuscript 

Response to reviewer [1]

Dear reviewer #1, thank you for your constructive comments for the betterment of our paper

1. Background

Line 74 to 76 needs a reference. 

Authors response; references are added in the revised manuscript 

The sentence starting from line 76 to 79 is not clear and need to be revised. 

Authors response; revised and written in readable/in clear way

2. Study setting

A brief discussion about the health care system in Ethiopia who give the reader insight.

Authors response; some details are included about the health care systems of Ethiopia in the revised manuscript.

3. Data source and study population

Line 117, the authors said the used the kid’s data set of EDHS 2016… my question why did you not use the women’s dataset and chose the kid’s dataset?

Author Response; For our study the Population base were “Women who have had one or more births in the 5 years preceding the survey” so we can use either KR (kids data set) or IR (women’s data set), the DHS manual/guide also indicates either of this can be used. Whether we use KR or IR data set there is no difference (give the same result). In KR file, the record is for the last birth only if we set Index to birth history (midx) = 1 but in IR dataset there were still unnecessary extra variables (for our study) such as midx_2, _midx_3, _..._6 , even though we can set midx_1=1, which makes us to extract variables somewhat tedious. 

4. Results

Line 174 to 176 you must be consistent with decimal places. 

Authors response; Corrected in the revised manuscript

Table 1 there were no values for parity. 

Authors response; Values are added in the revised manuscript

Line 177 to 180 needs to be revised. It is not clear. 

Authors response; we critically see it and amend accordingly in the revised manuscript to read as…... “Considering the characteristics of the community level factors” …...

Response to Reviewer [2]

Dear reviewer #2, thank you for your constructive comments you provided for the betterment of our paper

1. Lines 107-109. The author refers to the Ethiopian Census 2007 as the sampling frame of the EDHS 2016. This may not be very correct as DHS surveys only use the Census data to select Enumeration Areas. Then, a listing exercise is conducted in each selected EA to update the list of households. The updated list of hhds is used to select households to interview. The authors could revise the statement according to this methodology.

Authors response; we amended the sentence …... “The sampling frame used for the survey is the Ethiopia Population and Housing Census (PHC), which was conducted in 2007 by the Ethiopia Central Statistical Agency” …...to …… “To select enumeration areas for EDHS 2016, a total of 8,4915 Enumeration areas from an Ethiopian Population and Housing Census (PHC) conducted in 2007 were used as a sampling frame.” ….in the revised manuscript.

2. A conceptual framework to help readers understand why these variables were included in the study should have supported the description of the study variables

Authors response; A conceptual frame work is developed using different factors which are associated with delayed first ANC booking from different literatures by classifying into community and individual level (sociodemographic, pregnancy and maternal health related and behavioral factor) factors. 

3. Lines 145-147. Sample weights should be used throughout the analyses, not only for frequencies and for proportions. Could the author clarify whether the multilevel analysis was weighted or not.

Authors response; 

In the descriptive statistics we were used sample weighting to adjust disproportionate sampling, non-response or to restore representativeness. In addition, to get reliable standard error by taking in to account the sample design we calculate the confidence intervals for prevalence using svy set command (weighting for complex survey design) using svyset[pw=weight], psu(v021) strata(v023). Even though we applied a multilevel analysis to identify individual and community level factors by taking into account the clustering effect, the multilevel model cannot restore the representativeness of the data. Therefore, we apply sampling weight to restore the representativeness or non-response in the multilevel analysis also. In the revised manuscript we indicate as we done weighting throughout the analysis 

4. Were collinearity checks performed on the independent variables before ruining MV analyses?

Authors response; Even though, Stata is the robust software that automatically remove variables if there is multicollinearity, we also did multicollinearity test between independent variables by using pseudo linear regression analysis using the command “estat vif” and we got the mean VIF of 1.89. Therefore, there was no significant multicollinearity b/n predictor variables. 

5. The paragraph on the random effects does not add value to the paper; it rather brings statistical information that the author seems struggling to explain properly. The author uses values (va, π) that are not documented in the paper. Furthermore, I would expect the author to first clarify why a random effect model was needed on these data and then explain what those effects (if any) could be referring to in practice.

Authors response; we clarify why the random effect model was needed with their explanation by avoiding extra unnecessary descriptions in the revised manuscript. 

6. Presenting the results section by random vs. fixed effects seems to turn the paper to a methodological paper, comparing methods, rather than keeping the focus on describing the factors associated with delayed ANC. In addition, a number of statements in this section are subjective interpretations of statistical models which are not the focus of the paper and not expected in a result section. This section should be entirely rewritten to clearly highly the key results of the analysis that would be discussed later.

Authors response; Even though we remove the random effects part and put only the fixed effect result in a section/title “Individual and community-level factors associated with delayed first ANC booking” (by avoiding sections of random and fixed effect model), in the revised manuscript we critically modify/amend random effect and model fitness part and incorporate in the result section to convince or tell the reader how we were proceed with the multilevel model in order to identify factors associated with delayed first ANC booking, by considering the random effect and model fitness test. But if it does not make sense for you still, we can remove it and rearrange it accordingly.

---

## [Editor Report · Decision Letter 1]

9 Jun 2020

PONE-D-19-33776R1

Prevalence and Associated Factors of Delayed First Antenatal Care Booking among Reproductive Age Women in Ethiopia; A multilevel analysis of EDHS 2016 data

PLOS ONE

Dear Mr. Teshale,

Thank you for submitting your manuscript to PLOS ONE. After careful consideration, we feel that it has merit but does not fully meet PLOS ONE’s publication criteria as it currently stands. Therefore, we invite you to submit a revised version of the manuscript that addresses the points raised during the review process.

We would appreciate receiving your revised manuscript by Jul 24 2020 11:59PM. To enhance the reproducibility of your results, we recommend that if applicable you deposit your laboratory protocols in protocols.io, where a protocol can be assigned its own identifier (DOI) such that it can be cited independently in the future. For instructions see: http://journals.plos.org/plosone/s/submission-guidelines#loc-laboratory-protocols

We look forward to receiving your revised manuscript.

Kind regards,

Joshua Amo-Adjei, Ph.D

Academic Editor

PLOS ONE

Journal Requirements:

We noticed you have not corrected several occurrences of overlapping text with previous publications, which needs to be addressed.

In your revision ensure you cite all your sources (including your own works), and quote or rephrase any duplicated text outside the Methods section. Further consideration is dependent on these concerns being addressed.

Nancy Beam, PhD

Staff Editor

PLOS ONE

---

## [Author Response · Author response to Decision Letter 1]

13 Jun 2020

June 13, 2020 

Response to editor/reviewers

Title: Prevalence and Associated Factors of Delayed First Antenatal Care Booking among Reproductive Age Women in Ethiopia; A multilevel analysis of EDHS 2016 data

Manuscript number: PONE-D-19-33776R1

Editors comment

We noticed you have not corrected several occurrences of overlapping text with previous publications, which needs to be addressed.

In your revision ensure you cite all your sources (including your own works), and quote or rephrase any duplicated text outside the Methods section. Further consideration is dependent on these concerns being addressed.

Authors response: Dear editor thank you for your concern. We have checked our paper/manuscript in advance and we corrected text overlaps.

---

## [Editor Report · Decision Letter 2]

18 Jun 2020

Prevalence and Associated Factors of Delayed First Antenatal Care Booking among Reproductive Age Women in Ethiopia; A multilevel analysis of EDHS 2016 data

PONE-D-19-33776R2

Dear Dr. Teshale,

We’re pleased to inform you that your manuscript has been judged scientifically suitable for publication and will be formally accepted for publication once it meets all outstanding technical requirements.

Kind regards,

Joshua Amo-Adjei, Ph.D

Academic Editor

PLOS ONE
---

## [Editor Report · Acceptance letter]

23 Jun 2020

PONE-D-19-33776R2 

Prevalence and Associated Factors of Delayed First Antenatal Care Booking among Reproductive Age Women in Ethiopia; A multilevel analysis of EDHS 2016 data 

Dear Dr. Teshale:

I'm pleased to inform you that your manuscript has been deemed suitable for publication in PLOS ONE. Congratulations! Your manuscript is now with our production department. 

Kind regards, 

on behalf of

Dr. Joshua Amo-Adjei 

Academic Editor

PLOS ONE